# From Dietary Cholesterol to Blood Cholesterol, Physiological Lipid Fluxes, and Cholesterol Homeostasis

**DOI:** 10.3390/nu14081643

**Published:** 2022-04-14

**Authors:** Frans Stellaard

**Affiliations:** 1Department of Nutrition and Movement Sciences, NUTRIM (School of Nutrition and Translational Research in Metabolism), Maastricht University Medical Center, P.O. Box 5800 Maastricht, The Netherlands; f.stellaard@maastrichtuniversity.nl; 2Institute of Clinical Chemistry and Clinical Pharmacology, University Hospital Bonn, Venusberg-Campus 1, 53127 Bonn, Germany

**Keywords:** cholesterol, synthesis, absorption, bile acids, bile, hepatic, extrahepatic, lipoproteins, plant sterols, intestine

## Abstract

Dietary cholesterol (C) is a major contributor to the endogenous C pool, and it affects the serum concentration of total C, particularly the low-density lipoprotein cholesterol (LDL-C). A high serum concentration of LDL-C is associated with an increased risk for atherosclerosis and cardiovascular diseases. This concentration is dependent on hepatic C metabolism creating a balance between C input (absorption and synthesis) and C elimination (conversion to bile acids and fecal excretion). The daily C absorption rate is determined by dietary C intake, biliary C secretion, direct trans-intestinal C excretion (TICE), and the fractional C absorption rate. Hepatic C metabolism coordinates C fluxes entering the liver via chylomicron remnants (CMR), LDL, high-density lipoproteins (HDL), hepatic C synthesis, and those leaving the liver via very low-density lipoproteins (VLDL), biliary secretion, and bile acid synthesis. The knowns and the unknowns of this C homeostasis are discussed.

## 1. Introduction

High dietary cholesterol (C) intake is considered unhealthy, since it may contribute to elevated concentrations of serum C and in particular of low-density lipoprotein C (LDL-C). This is associated with the development of atherosclerosis and cardiovascular events, such as heart attacks and strokes. In this review paper, attention will be paid to the potential role of dietary C in the control of serum C. Serum C is the balance between the input of C into the blood and the C efflux from blood via hepatic extraction and elimination. The main C fluxes are shown in Figure 1, and they will be discussed in detail hereafter. In this introduction, only a generally accepted short summary is described. Consumed animal fat contains C in both the free (FC) and esterified form (CE). Fat digestion is predominantly established by the activated release of pancreatic digestive enzymes after meal intake and their secretion into the small intestine. Triglycerides (TG) are cleaved to free fatty acids (FFA) and glycerol, and C-esters to free C and fatty acids. In parallel, bile is secreted due to gallbladder contraction, activated by cholecystokinin (CCK) release from the duodenal mucosa. Bile acids (BA) enable the establishment of mixed micelles in which the hydrophobic FFAs and C as well as xenosterols such as plant sterols can be transported to the active sites of absorption. Here, the compounds are released and taken up into the enterocytes where they are predominantly incorporated into chylomicrons (CM) after esterification. After release of CMs into the lymph system, conversion into chylomicron remnants (CMR) starts through interaction with Lipoprotein Lipase (LPL), which converts TG to FFA. The FFAs are released from CM and the CMRs are much smaller and denser as CMs and taken up into the liver and potentially also into extrahepatic tissues and macrophages. Besides oxidized LDL, CMRs are deposited in coronary artery plaques. The liver is the central organ to control C homeostasis. Interrelated processes occur in the liver. Extracted with the CMRs, TGs and a part of CEs are released and incorporated into very low-density lipoprotein (VLDL) particles, which are secreted into blood. Through LPL, VLDL is converted to LDL particles that are re-extracted by the liver or by extrahepatic tissues such as macrophages. The C mixes with the hepatic C pool from which molecules can be used for BA synthesis and for biliary C secretion. The liver also takes up high-density lipoprotein (HDL) particles or the C component from the HDL particles from blood that are formed in extrahepatic tissues. This source of C is mainly used for biliary C secretion [1,2]. In bile, C is incorporated into mixed micelles formed by BAs and phospholipids. Bile is largely stored in the gallbladder at night and between meals. After meal intake, biliary C mixes with dietary C. Thus, dietary C is only part of the total C flux moving through the intestine before absorption. The daily flux of biliary C through the intestine is 2–4 times larger than the flux of dietary C [3,4]. The input of C into the whole-body C pool originates from C absorption and whole-body C synthesis; C is synthesized by almost all cells, and it is eliminated from the body via BA synthesis and fecal excretion of C and its bacterial metabolites coprostanol and coprostanone. In health, input and elimination are in balance leading to a constant whole body C pool. Imbalance caused by excessive input or reduced elimination may lead to elevated serum C. Ideally, high C absorption is compensated by a reduced in vivo C synthesis and an enhanced BA synthesis. Vice versa, enhanced C synthesis compensates for extremely lowered C absorption.

## 2. Dietary C Intake

In food, C is present in animal fat in the free form and predominantly in the esterified form. Animal fat is present in meat, fish, eggs, and in dairy products. In the western diet, daily C intake varies from approximately 100 to 400 mg/day, but excessive intakes up to 800 mg/day have been described [5]. C intake is much lower than the daily TGs intake of 60 to 150 g/day. In meat and fish, 100 g generally contains 40 to 120 mg of C. Restriction of C intake reduces the contribution to the daily intestinal C flux. Compared to omnivores, vegans have a 90% lower C intake, which results in a 13% lower serum LDL-C concentration [4]. This seemingly small effect on LDL-C is caused by a 35% increase in whole body C synthesis. This highlights the balance between absorption and synthesis. Therefore, a reduction in dietary C intake alone is not sufficient to lower serum LDL-C to a larger extent as needed in some diseased conditions. In addition, the balance is important in situations in which C intake is high. In that case, reduction of C synthesis keeps LDL-C under control. A convincing example is described by Kern Jr [6]. The daily diet of an 88-year old single man consisted of 25 eggs cooked in the morning and consumed during the day. His serum C was normal due to a reduced fractional absorption rate (FAR) and synthesis rate as well as an increased BA synthesis. Apparently, this compensating control does not function effectively in all human beings. Egg consumption is an ongoing discussion. One egg contains 150 to 180 g of C in the yolk fraction, which is relatively high as compared to the average daily intake of 100 to 400 mg C. The increased risk of cardiovascular events associated with a higher egg intake may be linked to the accompanying food products. With breakfast, eggs may be consumed fried in butter with bacon and sausages or consumed cooked with bread. Chinese populations tend to eat eggs with diner together with vegetables and rice. Interestingly, the risk among the Chinese for all-cause mortality does not increase with increasing egg consumption up to 7 eggs/week [5]. This was in contrast with black and white Americans whose risk increases already from 3 eggs/week on. Moreover, egg intake has been shown to increase HDL-C in serum in parallel with LDL-C, which may partly compensate for the risk effect of a high LDL-C concentration. A recent meta-analysis by Kuang et al. [7] indicates that the unchanged HDL-C/LDL-C ratio may explain that egg consumption does not increase the risk for cardiovascular disease in the general population. Patients at risk for cardiovascular disease should limit egg consumption. In another recent review paper [8], the authors discuss the recommendations on egg consumption for patients at risk for cardiovascular disease. They conclude that the rate of C absorption is important. Indeed, the FAR varies individually from 20 to 80% [9,10]. A healthy subject consuming 300 mg C per day and absorbing 80%, absorbs 240 mg C per day. A healthy subject consuming 100 mg while absorbing 20%, absorbs only 20 mg. The difference in absorptive load of dietary C between the two subjects is 220 mg per day or 12-fold. The FAR affects not only dietary C but also biliary C. Thus, a high FAR may lead to elevated serum total C and LDL-C concentrations. 

## 3. Known and Unknown C Fluxes in Humans

In Figure 1, average sizes of known C fluxes are indicated. The numbers are retrieved from the data for omnivores as published by Lütjohann et al. [4]. With a mean daily C intake of 350 mg/d, the subjects had a relatively high C consumption, when compared to the normal intake of 100 to 400 mg/d. The data show that the daily biliary C flux is twice the dietary C flux (700 mg/d). The total daily intestinal C flux is then 1050 mg/d. Based on the mean 50% FAR, the daily total flux of absorbed C is on average 525 mg/d. The majority will be taken up by the liver, the rest enters extrahepatic tissues including macrophages. The exact proportions are not known. We assumed 450 mg/d to be extracted by the liver. The measured whole-body C and bile acid synthesis rates were 840 and 420 mg/d. The exact contribution of hepatic C synthesis to whole-body synthesis in humans is not known. In the cynomolgus monkey, this value is around 20%, which may represent the value in humans [11]. Generally, in scholarly literature, a 50% contribution is assumed without references. Both options would lead to a hepatic synthesis of 165 and 420 mg/d. An in-between value of 300 mg/d was chosen as a compromise. Despite the clear high uncertainty, this value gives a rough estimate for hepatic synthesis. It must be realized that large inter-individual variations exist in dietary C intake, biliary C secretion, C FAR, and BA synthesis. The whole-body C synthesis compensates for the fecal loss of endogenous C and BA synthesis. The total influx of CMR-C into the liver and hepatic C synthesis together estimates 750 mg/d, much less than the sum of biliary C secretion rate of 700 mg/d and BA synthesis of 420 mg/d. The flux of C entering the liver in HDL particles may largely contribute to biliary C secretion. The hepatic LDL-C influx is smaller than the VLDL-C efflux. Daily flux values for the hepatic VLDL-C secretion, LDL-C uptake, and HDL-C uptake are not available. The VLDL secretion is mainly controlled by the triglyceride flux entering the liver with CMRs. The hepatic LDL uptake is mainly determined by VLDL secretion, fatty-acid release from VLDL, hepatic uptake of VLDL remnants, uptake of LDL into extrahepatic tissues, and hepatic LDL-receptor activity. Furthermore, biliary secretion is here defined as the daily flux of biliary C entering the intestine. This flux contains mostly C derived from bile stored in the gallbladder which is emptied several times a day. Intestinal biliary C is partly absorbed and recycled to the liver. Therefore, the amount of hepatic C newly secreted into bile is much lower than the flux of biliary C entering the intestine. The daily biliary C secretion rate into the intestine is dependent on the degree of gallbladder contraction, which may be 50 to 90% [12] and the number of gallbladder contractions being two to three per meal intake. The numbers depend on subject, meal size, and meal composition (fat and protein content). In case of extreme high fat meals, gastric emptying is delayed and the gallbladder may remain in a contracted state for many hours keeping the bile in continuous cycling. Thus, C homeostasis requires control over a complex system of fluxes.

Another largely unexplored aspect of C metabolism is the bacterial metabolism of C that escapes small intestinal absorption. Based on the fluxes presented in Figure 1, the daily flux of malabsorbed C averages 525 mg/d. The major metabolic metabolite is coprostanol [13]. The conversion rate, expressed as the coprostanol/cholesterol ratio in the feces of healthy subjects is high (high converters), but may be low in a minor group of subjects (low converters). Only a small number of bacterial strains has been identified that is able to produce coprostanol from cholesterol. This process has been shown to modify host C levels in serum [14]. The ratio of coprostanol/C in feces appears inversely related to the serum C concentration [15]. Diet factors may intervene with the colonic bacterial C metabolism as has been documented for milk polar lipids, reducing serum C concentration [16]. 

## 4. Fat Digestion and Intestinal Transport

The dominant lipids in the diet are TGs. In the western diet, the fat intake predominantly consisting of TG is between 60 and 150 g/day. The C intake comprises 100–400 mg/day. Considering an average TG intake of 100 g/d and a C intake of 200 mg/d, the TG/C ratio is 500. The digestion of TG and C require gastric emulsification and de-esterification with gastric but predominantly pancreatic lipases and cholesterol esterase and uptake in mixed micelles formed by BAs (Figure 2). The availability of BAs is determined by liver function and gallbladder motility. During gastric emulsification, fat is divided in droplets being slowly reduced to micro-droplets. These allow access to digestive enzymes and bile after their entrance into the duodenum. The appearance of fat in the duodenum leads to CCK release from the duodenal cells. The CCK stimulates secretion of pancreatic juice containing digestive enzymes and the activation of gallbladder contraction. Pancreatic juice and bile migrate to the duodenum. TGs are broken down to FFA, glycerol and monoacylglycerol, CE to C, and fatty acids. BAs initiate the formation of mixed micelles in which fatty acids and C can be packed for transportation through the small intestine. Conjugated plant sterols such as sitosterol and campesterol are also present in food and transported into the micelles in its free form. The daily intake of plant sterols is similar to the intake of C. It is generally assumed that in healthy subjects emulsification, de-esterification and micellar uptake are always highly efficient for TG and C. It is also assumed that the large load of TGs does not impair the processing of C. The micelles containing FFA and C move down the small intestine and FFA and C are delivered to the enterocytes, where they are taken up by specific transporter proteins.

## 5. Absorption

The micelles allow lipids to be transported through the aqueous environment, to cross the unstirred water layer and to attach to the enterocyte in order to release free fatty acids and sterols into the cell (Figure 2); C is predominantly absorbed in the jejunum [17,18]. For sterols, Niemann-Pick C1 Like 1 (NPC1L1) is the selective uptake protein [19]. The enterocytes esterify free fatty acids and sterols back to triglycerides and sterol esters before incorporation into large lipid rich CMs carried by the ApoB-48 lipoprotein (Figure 3). After structuring and lipid loading of the chylomicron particles, they are secreted into the lymph. There are indications that C may be partly transported out of the enterocyte directly into blood via HDL [20,21]; CMs are transported through blood. A protein system consisting of ATP-binding cassette sub-family G member 5 and 8 (ABCG5/G8) has been shown to be effective in re-secreting part of the already taken up C and plant sterols back into the intestinal lumen [22,23]. The effect is larger on plant sterols. This may be explained by the smaller degree of esterification of plant sterols in the enterocyte. The overall FAR of TG in healthy subjects is high, i.e., >85%, mean values of >90% have been described [24,25]. In contrast, the overall FAR of C in healthy subjects is highly variable (20–80%), with a mean value of approximately 50% [9,10]. The large variation in the FAR of C is generally ascribed to the variable activities of NPC1L1 and ABCG5/G8. It has been shown that the whole length of the small intestine contains cells that enable absorption of the large FFA load. The smaller load of C may need a smaller area. The duodenum and jejunum are generally indicated as the regions for C absorption. However, whether FFA and C are dissociated in micelles is not known. On average, healthy subjects have an orocecal transit time of 4–6 h after consuming a solid meal. Part of this transit time is determined by gastric emptying, which is delayed when the food is enriched in fat. Reduced activity of intestinal ABCG5/G8 induced by gene mutations leads to enhanced absorption of C and plant sterols as shown in sitosterolemia patients [26,27]. The FARs for plant sterols are below 20% [28], which explains their low serum concentrations. Their low absorption rates are mainly due to the low intestinal re-esterification rate which makes the plant sterols more susceptible for re-secretion into the lumen by ABCG5/G8. 

Both NPC1L1 and ABCG5/G8 are also involved in hepatic C secretion into bile [29,30]: NPC1L1 controls re-absorption of C from bile over the canalicular membrane back into the liver, and ABCG5/G8 regulates the C secretion into bile. General whole-body upregulation of ABCG5/G8 would enhance biliary C secretion, reduce absorption, and increase fecal C excretion, i.e., increased elimination. Upregulation of ABCG5/G8 will have a little effect on plant sterol absorption, but possibly a moderate effect on dietary C absorption. However, the combination with increased biliary C secretion may positively affect total C elimination. Plant stanols and sterols are being used as food supplements in order to lower C absorption [31,32]. This effect of plant sterols is generally ascribed to a competitive effect for the inclusion into mixed micelles. Plant sterols reduce the uptake of C into the micelles [33]. 

Chylomicron formation, loading and release appears to be a slow process. The TG produced at the level of the endoplasmic reticulum (ER) is either incorporated into pre-chylomicrons within the ER lumen or shunted to TG storage pools [34,35]. This protects the body against the large load of TG introduced by a meal. The absorption is spread out over the whole day including the fasting state. Isotope studies show that the appearance in blood of an orally administered bolus of labeled TG maximizes after approximately 8 to 12 h [24]. Oral administration of isotopically labeled C results in a maximum isotope enrichment in plasma after approximately 24 h [36]. The CMR formation kinetics is determined by the supply of CM and by LPL activity. In health, LPL activity is normally high, so that CMR formation is predominantly determined by the CM supply. It must be understood that CM supply and conversion of CM to CMR are determined by TG, not by C; C is just transported by CM and remains transported in CMR.

## 6. Hepatic C Metabolism

The CMRs that reach the liver are taken up by the LDL-receptor-related protein (LRP), and TG and CE are released (Figure 4). It may be predicted that 100 g TG and 200 mg C are consumed per day, that 400 mg/d biliary C is added, that 95% of TG and 50% of C are absorbed and that 10% of absorbed TG and 90% of absorbed C reach the liver. In that case 9.5 g TG and 270 mg C enter the liver per day with a TG/C ratio of approximately 35. To prevent fatty liver disease, TG is removed by incorporation into VLDL particles. In fasting serum VLDL, the average physiological TG/C ratio equals approximately 5 (*w*/*w*), which means that additional C is secreted and/or that part of TG in VLDL has already been de-esterified and released as free fatty acids at the moment of sample collection. Furthermore, the predictions assume that absorbed TG and C leave the enterocyte at the same time and with the same speed. This may not be true. The speed of incorporation of TG and C into chylomicrons may not be the same. The C in the liver, originating from CMR, may be secreted in VLDL, converted to bile acids or secreted into bile. The same may happen to *de novo* synthesized C in the liver and C that returned to the liver via LDL. The control over the distribution of these fluxes is largely unknown. 

C is synthesized by a complex enzyme system. The rate limiting enzyme involved is 3-hydroxy-3-methylglutaryl coenzyme A reductase (HMG-CoA reductase). BAs are solely synthesized in the liver and circulate in the enterohepatic circulation with a very high FAR (~95%) in the terminal ileum. Two important pathways convert C to BAs [37]. The dominant one (neutral pathway, ~75%) involves 7α-hydroxylation of C as the first and rate limiting step. The second one (acidic pathway, ~25%) starts with 27-hydroxylation of C. BA synthesis has its own control. In the ileocyte the Farnesoid X receptor (FXR) is activated by BAs which leads to release of fibroblast growth factor 19 (FGF19) which inhibits the rate limiting enzyme in BA synthesis C-7α-hydroxylase. Decreased BA absorption induced by colesevelam and cholestyramine enhances BA synthesis and as a consequence hepatic C synthesis. Inhibited hepatic C synthesis induced by statins, does not reduce BA synthesis [38]. C synthesis and BA synthesis undergo diurnal variation [39,40,41,42]. C synthesis is predominantly stimulated at night, whereas BA synthesis is activated in the early evening.

## 7. Biliary C Secretion and Trans-Intestinal C Excretion (TICE)

In addition to BA synthesis, biliary C secretion is an approach to remove C from the body. It is unclear how biliary C secretion is controlled. Is C released from HDL its only substrate or is an increased hepatic C pool size an activator for induction? Gallbladder contraction creates an enhanced BA flux through the intestine and liver that is automatically forwarded toward the gallbladder. Also, biliary C is partly absorbed and recycled back to the liver. However, with an intestinal transit time of approximately 30 to 60 min, cycling of BAs is very rapid compared to the cycling of C [43]. Transport of absorbed C is strongly delayed by chylomicron metabolism. This may mean that during cycling after a meal, the bile becomes depleted in C and enriched in BAs. On the other hand, cycling BAs may have a stimulatory effect on hepatic C secretion into bile. In humans, measurement of biliary C secretion requires very invasive techniques applying triple lumen intubation into the small intestine. The alternative technique is to determine dietary C intake, fecal excretion of C metabolites and BAs (sterol balance method) and determination of the C FAR. This allows calculation of total C loss and dietary C loss. The difference reflects biliary C loss. In mouse models, it was possible to estimate biliary BA and C secretion by puncturing the gallbladder after occlusion of the bile duct and collecting the bile secreted by the liver. It was observed that this flow was smaller than was calculated by the fecal balance method. An alternative route of C removal was defined as the trans-intestinal C excretion (TICE) to complete the balance [44,45]. In a later stage, a new method was developed to estimate biliary C secretion in humans since quantitative bile collection is not possible in humans; BA kinetics were measured with stable isotopes as well as the fractional turnover rate of BAs in plasma. These data were transferred to a biliary BA secretion rate. In bile, collected with CCK induced gallbladder contraction, biliary C secretion rate was calculated. Using this procedure also in humans a TICE flux could be demonstrated [36]. Intestinal ABCG5/G8 is most likely involved in removal of TICE C from the enterocyte. In parallel with biliary and dietary C, also C directly secreted into the intestinal lumen is subject to reabsorption [46]. Interestingly, this indicates incorporation of TICE C into micelles passing the small intestine, which makes the understanding of C homeostasis more complex. Otherwise, for balance studies, biliary C excretion and TICE do not necessarily need to be separated. In balance studies a total fecal C excretion rate is measured independent of the source. After correction for loss of dietary C, a value remains for loss of biliary C and TCE together. 

## 8. Lipoprotein Metabolism

The hepatic metabolism of CMs and CMRs have been discussed in the previous sections. VLDL particles are secreted from the liver into blood. As for CMs, TGs in VLDL are de-esterified by LPL and FFAs are released and stored in adipocytes or muscles. The VLDL particles decrease in size and density and gradually turn over into VLDL-remnant particles of which LDL particles are the smallest. LDL and partly also VLDL remnants can be taken up by the liver, but also by extrahepatic tissues. The difference between VLDL-remnants and LDL is no longer strict. In addition to LDL-C, VLDL remnants C attribute to the risk for cardiovascular disease [47,48]. The terminology of non-HDL-C becomes popular and stands for all ApoB-100 lipoprotein bound C, i.e., LDL, VLDL and VLDL remnants [49]. In parallel, there is a trend to promote the measurement of ApoB-100 as a more valuable marker for atherosclerotic risk [50,51]. This is based on the knowledge that not only the C concentration inside lipoprotein fraction is important, but also the number of and the size of particles. VLDL and LDL consist of particles with different sizes. The more abundant and the smaller the LDL particles, the more atherogenic they are [52,53,54]. Independent of size, each particle contains one ApoB-100 unit. HDL transfers C from extrahepatic sites including macrophages and plaques to be delivered to the liver and secreted into bile for removal. However, HDL particles and LDL particles do not act independently. It has been demonstrated that in blood, CE can be transferred from HDL to VLDL and LDL by the cholesterol ester transfer protein (CETP) [55]. Thus, a high CETP activity lowers HDL-C and increases LDL-C. A low activity does the opposite, and it is protective against atherosclerosis. Medication to lower CETP is still under development, but it has not been successful so far [56,57]. 

## 9. C Flux Measurements in Humans

The first and principal C fluxes are synthesis and absorption. Whole body C synthesis is measured using the sterol balance method in which fecal C excretion, fecal BA excretion, and dietary C intake are measured [58]. Alternatively, C synthesis is measured via the mass isotopomer distribution analysis (MIDA) [59] applying stable isotopes. As C is a polymer of acetate units, infusion of ^13^C-acetate leads to a continuously increasing isotope enrichment of the C pool. The increment of enrichment in serum C reflects the fractional C synthesis rate. In parallel, a methodology was developed employing deuterated water as the substrate [60]. Both the sterol balance method and isotope dilution techniques are not suitable for C synthesis measurement in large patient populations or patient care. For that purpose, surrogate marker technology was developed measuring intermediate metabolites in the C synthesis pathway. Squalene, lanosterol, desmosterol, and lathosterol have been studied, but the lathosterol concentration is the most generally accepted marker [61]. Stable isotope technology is used to measure C absorption; ^13^C labeled or deuterated C is administered orally incorporated in a meal or a capsule during a meal. Two approaches have been introduced. One applies a seven-day continuous isotope feeding protocol combined with labeled sitostanol as minimal absorbable reference sterol and a three-day fecal collection period. From the ratio of enrichments of both labeled sterols in feces collected during the last three days, the C FAR can be calculated [9]. The second approach applies a single oral dosage of labeled C combined with a simultaneous intravenous dosage of a different labeled C. The ratio of both isotope enrichments in serum two to three days after administration allows the calculation of the C FAR [10]. Both approaches are unsuitable for studies in large patient populations or in patient care. Also, for C absorption surrogate markers have been evaluated. Cholestanol and various plant sterols have been studied of which the serum concentrations of campesterol and sitosterol are mostly used [62]. Plant sterols are much less efficient absorbed [63], but a high plant sterol absorption is associated with a high C absorption. Finally, BA synthesis is measured using the stable isotope dilution method administering ^13^C or deuterated primary bile acids (cholic acid, chenodeoxycholic acid) in a single oral bolus and measuring the decay of isotope enrichment in serum during four days [64,65]. Alternatively, BA synthesis is measured as the rate of fecal excretion, since BAs escaping absorption are lost via feces as primary or secondary BAs. The concentration in serum of the intermediates in the BA synthesis pathway 7α-hydroxycholesterol and 7α-hydroxy-4-cholesten-3-one are used as markers for BA synthesis [66,67]. Generally, marker concentrations in serum are corrected for the total C concentration in order to correct for variable lipoprotein metabolism and LDL-receptor activity. It must be realized that a correlation between a marker/C ratio in a single fasting serum sample and the function measured over a four or a seven day period contains large variation. Thus, markers are valid for group comparisons but less valid for individual patients. Biliary C secretion cannot be measured directly in humans. Generally, it is calculated in the situation that fecal C excretion, dietary C intake, and the C FAR have been measured. The difference between total fecal C excretion and dietary C excretion is interpreted as the loss of biliary C. Via FAR, the biliary secretion may be calculated as discussed in Section 7. For C absorption, it may need discussion what flux is important. The measurement of C absorption presents a value for the C FAR, i.e., the percentage of the dosed C isotope that is absorbed. Important additional parameters are the dietary C intake and the biliary C secretion rate. Together with FAR they determine the daily fluxes of absorbed dietary and absorbed biliary C as well as the total daily C absorption rate. Dietary C intake is normally determined via a nutrition diary. Different research centers may use different information on C contents of food products and different protocols. Diaries also tend to underestimate food intake. The fluxes of C in the serum lipoproteins VLDL, LDL and HDL cannot be measured. The turnover of these lipoproteins has been measured by collecting the lipoproteins from blood, labeling these with isotopes, reinjecting these into the volunteer and measuring the disappearance from blood [68,69]. These techniques are only experimental, and they depend largely on extensive modeling. No alternative marker techniques are available. For patient care in the hospital so far only the classical parameters serum total C, LDL-C, HDL-C are available. In the near future the parameters serum non HDL-C and ApoB may be included in the selection of patients to be treated for prevention of cardiovascular disease. The surrogate markers for C absorption and C synthesis might be introduced in order to predict whether a patient is a high C absorber or high C synthesizer [70]. However, the validity of markers for this purpose may be limited. 

## 10. Discussion

C homeostasis is the process that balances C input and C elimination. In detail, this includes all the steps in C metabolism as discussed above, involving transport of absorbed C in CMs and CMRs, endogenously synthesized C in hepatic and extrahepatic tissues, transport of C in VLDL, LDL and HDL particles, biliary C secretion and TICE as well as synthesis of BAs. Each step requires its own research. Many details are being unraveled at the cellular level discovering molecular mechanisms involved. Updates of information obtained in molecular mechanisms can be found in the recent review papers by Luo and Li [71,72]. Here we focused on the C fluxes in healthy conditions. The interactions between the different steps are so far largely unknown. Generally spoken, C absorption and C synthesis are balancing in that synthesis adjusts to changes in absorption and vice versa. A subject’s absorption rate is determined by many factors as described above. Summarized they are:C intake;Digestion including gastric function, pancreatic function;Hepatic BA synthesis and biliary secretion;Biliary C secretion (NPC1-L1, ABCG5/G8);Cholecystokinin (CCK) production and release;Gallbladder function (contraction in response to CCK, relaxation);Intestinal micelle formation, micellar uptake of C;C uptake into enterocytes (NPC1-L1) and re-secretion into intestinal lumen (ABCG5/G8);TICE;CM formation, release and conversion to CMRs.

Every step involved has its own individual efficiency contributing to the daily flux of C entering the body C pool. The body C pool must sense excess or shortage of C being absorbed in order to upregulate or downregulate C synthesis. Extrahepatic tissues are able to synthesize C, take up C from blood as LDL particles, but they are largely unable to metabolize C. Their excess C must be removed via HDL transport. The liver is the intermediate in C homeostasis. Strongly reduced C absorption will reduce the hepatic C pool and stimulate hepatic C synthesis and increase the LDL-receptor activity stimulating C uptake. Excess C absorption may result into enhancement of the hepatic C pool, reduced hepatic C synthesis, enhanced BA synthesis and biliary C secretion but also enhanced VLDL secretion and LDL formation. In the ideal situation, serum C concentration does not increase. However, the hepatic responsiveness to strongly enhanced absorption may be impaired leading to increased serum LDL-C and total C concentrations. A reduction in the flux of absorbed C can be created by a reduction in dietary C intake and a reduction in C absorption. The latter option is achieved by pharmacological action of ezetimibe that reduces the intestinal NPC1L1 activity or by increased dietary intake of plant sterols or stanols as food supplements that compete with C uptake in micelles. However, plant sterols and stanols as food additives are under discussion [73,74]. A major drawback of reducing the C absorption rate remains compensation by increased synthesis. This means that the net effect is the difference between the reduced absorption and the increased compensatory synthesis. In order to highly efficiently lower serum C levels, reduction of absorption must be combined with a means to reduce C synthesis. In hypercholesterolemic patients, statin treatment is the first choice to lower serum C. In case of insufficient serum C reduction, ezetimibe treatment is added as a co-treatment. So far, only statins are available as the basic pharmacological tool to reduce C synthesis. For patients that do not tolerate statins, Bempedoic acid can now be prescribed, which blocks adenosine triphosphate-citrate lyase in the liver which is involved in C production [75]. When the combined treatment does not achieve sufficient LDL-C reduction, the next step is additional treatment with a proprotein convertase subtilisin/kexin type 9 (PCSK9) inhibitor [76]. The PCSK9 inhibitor binds to the LDL-receptor and inhibits receptor activity. This results in enhanced serum LDL-C levels. Two PCSK9 inhibitors are now on the market: Alirocumab and Evolocumab. Additionally, Inclisiran has been developed, a small interfering RNA that inhibits translation of the PCSK9 protein [77].

## 11. Conclusions

The C homeostasis process is complex and highly regulated, including hepatic and extrahepatic C synthesis, uptake and release processes, intestinal absorption, various lipoprotein transport systems in blood, biliary secretion, and trans-intestinal excretion as well as BA synthesis. The major pathological focus is on serum C and in particular LDL-C in relationship to cardiovascular disease. Reduced dietary intake and dietary means to reduce C absorption have been shown to be effective tools to reduce serum C. However, their efficacies are limited due the contribution of biliary C and to the compensatory induction of C synthesis.

## Figures and Tables

**Figure 1 nutrients-14-01643-f001:**
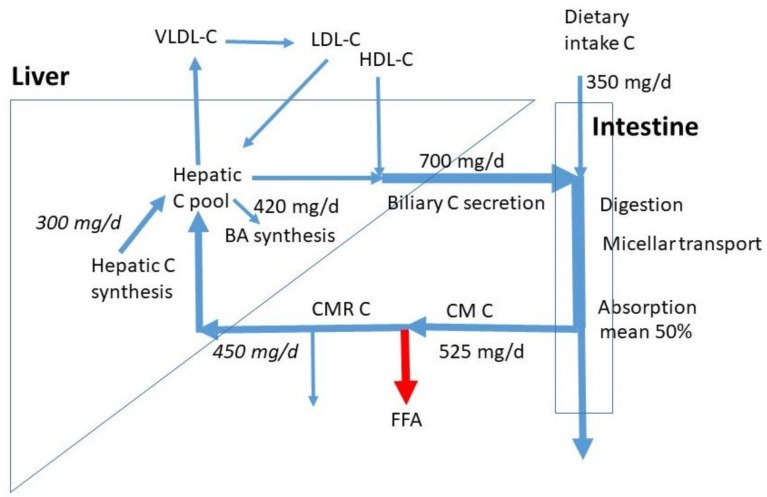
C fluxes in omnivore humans as published by Lütjohann et al. [4]. The numbers in italics are obtained by reasonable estimates based on the proportion of chylomicron remnants being extracted by the liver and the proportion of hepatic C synthesis to whole body synthesis. Chylomicron remnants not extracted by the liver are taken up by extrahepatic tissues including macrophages. CM = chylomicrons, CMR = chylomicron remnants, VLDL = very low density lipoprotein, LDL = low density lipoprotein, HDL = high density lipoprotein, FFA = free fatty acid, BA = bile acid. No numbers are available for the lipoprotein fluxes.

**Figure 2 nutrients-14-01643-f002:**
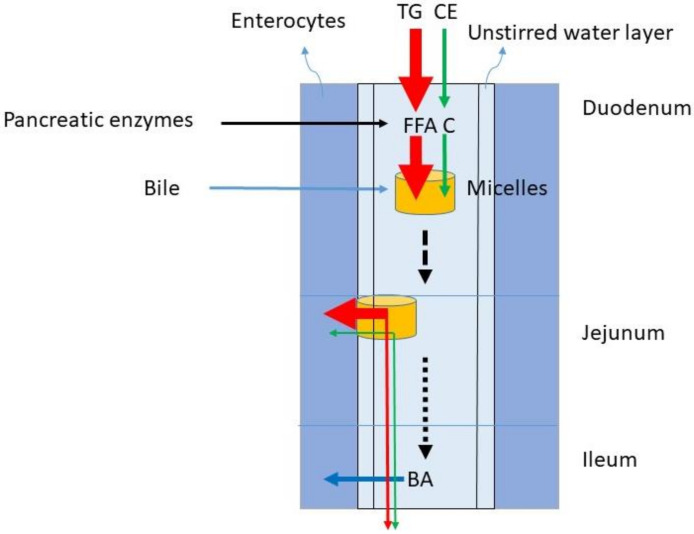
Fat digestion of TG and CE and micellar transport of FFAs and C to enable lipid uptake into the enterocytes. The red arrow represents TG metabolism, the green arrow C metabolism and the blue arrow BA absorption. TG = triglycerides, CE = cholesterol ester, FFA = free fatty acid, C = free cholesterol, BA = bile acid.

**Figure 3 nutrients-14-01643-f003:**
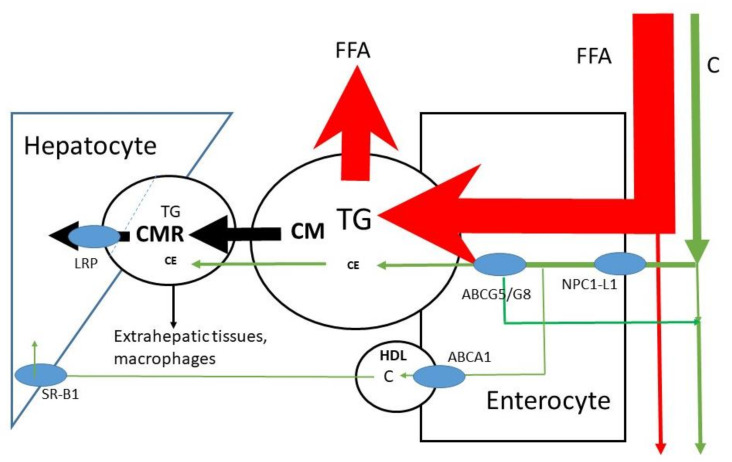
Uptake of FFA and C into enterocyte, CM secretion and conversion to CMR and uptake of CMR into the liver. Partial transport via HDL is also indicated. HDL= high density lipoprotein. TG = triglyceride, FFA = free fatty acid, C = free cholesterol, CE = cholesterol ester, CM = chylomicron, CMR = chylomicron remnant, ABCG= ATP-binding cassette sub-family G, NPC1L1 = Niemann–Pick C1 Like 1, ABCA1 = ATP-binding cassette transporter ABCA1, SR-B =Scavenger receptor class b type 1, LRP = LDL-receptor-related protein.

**Figure 4 nutrients-14-01643-f004:**
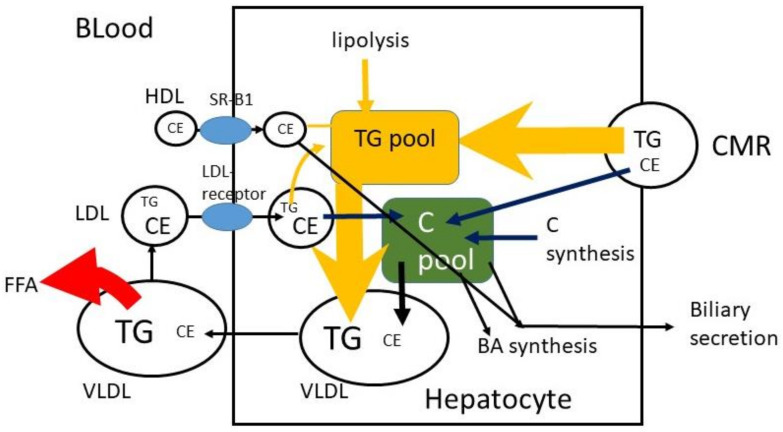
Hepatic C metabolism. TG = triglyceride, FFA = free fatty acid, C = free cholesterol, CE = cholesterol ester, CM = chylomicron, CMR = chylomicron remnant, BA = bile acid.

## Data Availability

Not applicable.

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
