# Peer review of "From Dietary Cholesterol to Blood Cholesterol, Physiological Lipid Fluxes, and Cholesterol Homeostasis"

_nutrients, 2022, doi:10.3390/nu14081643_

Round 1
Reviewer 1 Report
It is an extremely well written review with elaborate details on cholesterol flux and homeostasis and includes appropriate supporting figures for a clear understanding on how serum cholesterol levels are controlled by the liver metabolism.
1. It is ready to be accepted. I have a minor suggestion. It is a suggestion to the author if they want to include any more details on fatty liver and how dietary cholesterol could contribute to it directly vs the contribution to cardiovascular diseases.
Author Response
Reviewer 1
It is an extremely well written review with elaborate details on cholesterol flux and homeostasis and includes appropriate supporting figures for a clear understanding on how serum cholesterol levels are controlled by the liver metabolism.
- It is ready to be accepted. I have a minor suggestion. It is a suggestion to the author if they want to include any more details on fatty liver and how dietary cholesterol could contribute to it directly vs the contribution to cardiovascular diseases.
Response:
I am very grateful for the positive reply of the reviewer. An important aspect in writing this review was, that it is meant to serve as an introductory review for the special Nutrients issue “From dietary cholesterol to blood cholesterol”. This issue tries to link dietary cholesterol to blood cholesterol and as a consequence to cardiovascular disease. Therefore, in my opinion, the role of cholesterol to fatty liver seems outside the scope of the paper. Furthermore, due to the diverse hepatic fluxes of endogenous and exogenous cholesterol, the contribution of dietary cholesterol may be difficult to explain.
Reviewer 2 Report
Whole body cholesterol levels are carefully maintained by balancing input (intestinal absorption and cholesterol synthesis) and elimination (conversion to bile acids and faecal excretion). This is a complex area, involving multiple interlinked pathways with many of the underlying regulatory mechanisms still unknown. The area is hampered further by difficulties in technologies to measure and investigate pathways in humans. Cholesterol metabolism involves the intestine, blood, liver and gallbladder. This review describes each of the steps involved in cholesterol metabolism from uptake to excretion and summarizes the current knowledge of what drives the fluxes between the various steps, paying particular attention to the role of dietary cholesterol.
This manuscript gives a clear overview of the complexity of cholesterol metabolism, what is known and what is unknown in this area. The language, in some parts, is quite colloquial and many statements are general in nature and it is not clear whether they are thoughts of the author or whether they are based on scientific literature.
Main points
Figure 1 is not very clear. Most of the abbreviations are not explained and it is unclear what the thickness of the lines indicates. Why is the FFA line red? What does the blue line near CMR represent? There are no numbers indicated on the line for VLDL synthesis. If the flux for this is unknown this should be indicated.
Line 54: It is mentioned here and elsewhere that cholesterol derived from HDL is mainly used for biliary cholesterol secretion. Reference should be added
Line 58; It is stated that “The daily flux ….” Reference should be included
Lines 86-111: A whole section is devoted on the discussion about egg consumption, which seems out of place and distracts from the point the author is trying to convey. The point regarding the balance of uptake and synthesis and the importance of FAR has been made by describing Kern’s study. The next section is confusing diverts into accompanying foods, LDL and HDL levels. There is a statement suggesting that patients at risk for cardiovascular disease should limit their egg consumption and it is not clear what this is based on. Then the section moves back to FAR.
Line 82: ‘this proofs’ – colloquial (this highlights)
Line 86: ‘nice example’ – colloquial
Section 3: This section seems to be predominantly based on data from Lutjohann et al. Are there any confirmatory studies?
Section 3 contains a lot of individual statements with no references and it is not always clear how these statements are linked together. A table or a figure indicating what is known and not known maybe helpful
Line 115: ‘the subjects consumed relatively much cholesterol’ – what is this statement based on?
Line 118-119: ‘We assumed 450 mg/d to be extracted by the liver’ – based on which studies?
Line 122: ‘Generally, a 50% contribution is assumed’ – based on what? Reference?
Line 123: ‘As a compromise here a value of 300 mg/d is applied’ – by who and as a compromise to what?
Lines 126-131: Lots of statements with no references.
Line 129: How does LDL-C influx relate to CMR-C influx?
Line 190-191: ‘The time frame in which the micelles pass the duodenum and jejunum may be around two hours’ – add reference
Line 196: new paragraph
Lines 189-200 and 200-203 seem to state the same point
Line 206: new paragraph
Line 214: In health, rephrase
Figure 2: the figure suggests that absorption is only occurring in the jejunum
Line 231: The LDLr is not the only receptor involved in CMR uptake
Line 232: ‘Let’s predict’ – colloquial
Line 245: new paragraph and not sure why the HDL comment is inserted here as it has been mentioned twice before. Similarly, it is unclear why ‘C is synthesized as a polymer of acetate units. The rate limiting enzyme involved is 3-hydroxy-3-methylglutaryl coenzyme A reductase (HMG-CoA reductase)’ are mentioned here.
Line 252: it is unclear what is meant by ‘BA synthesis has its own control’
Figure 4: from the figure it appears that HDL-derived CE is contributing to the TG pool
Line 270: replace ‘otherwise’ with ‘on the other hand’
Lines 280-281: ‘In a later stage, a new method was developed to estimate biliary C secretion in humans’ - Provide more detail on how the method changed. It is unclear how this relates to the next sentence: ‘Also, in humans an undefined external flux added to faecal C excretion’
Lines 287-288: ‘Otherwise, for balance studies, biliary C excretion and TICE do not necessarily need to be separated’ – rephrase and explain
Line 291: remove also
Line 292: replace ‘is’ with ‘are’
Line 295: ‘Recently, the difference between VLDL-remnants and LDL is not strict anymore’ – colloquial, rephrase and add reference
Line 302: The more – colloquial (more abundant?)
Line 303: ‘Independent of size, each particle contains one ApoB-100 unit’ - it is unclear whether this only applies to LDL or also VLDL and VLDL-remnants
Line 315: duplication of line 245
Section 9: A table listing the different methods with pro’s and con’s would be helpful
Line 352: ‘For C absorption, it may need discussion what flux is important’ is unclear.
Line 376: The statement: ‘Each of the steps require its own research’ seems at odds with the main message of the manuscript, that cholesterol homeostasis is complex, involves many pathways that are interlinked. They should be studied in the context of each other.
Line 411: ‘anyway’ - colloquial
Line 415: it is stated that only statins are available for the treatment of hypercholesterolemia which is not correct. There are other treatment options such as PCSK9 inhibitors.
Minor point
Abbreviations not consistently used throughout the manuscript
Author Response
Reviewer 2
Whole body cholesterol levels are carefully maintained by balancing input (intestinal absorption and cholesterol synthesis) and elimination (conversion to bile acids and faecal excretion). This is a complex area, involving multiple interlinked pathways with many of the underlying regulatory mechanisms still unknown. The area is hampered further by difficulties in technologies to measure and investigate pathways in humans. Cholesterol metabolism involves the intestine, blood, liver and gallbladder. This review describes each of the steps involved in cholesterol metabolism from uptake to excretion and summarizes the current knowledge of what drives the fluxes between the various steps, paying particular attention to the role of dietary cholesterol.
This manuscript gives a clear overview of the complexity of cholesterol metabolism, what is known and what is unknown in this area. The language, in some parts, is quite colloquial and many statements are general in nature and it is not clear whether they are thoughts of the author or whether they are based on scientific literature.
Main points
Figure 1 is not very clear. Most of the abbreviations are not explained and it is unclear what the thickness of the lines indicates. Why is the FFA line red? What does the blue line near CMR represent? There are no numbers indicated on the line for VLDL synthesis. If the flux for this is unknown this should be indicated.
Response:
I thank the reviewer for pointing out some clear weaknesses in figure 1. All abbreviations are now clarified in the legend. The different thicknesses of the lines indicate the differences in flux sizes due to absorption and diversion of CMR into liver and macrophages as well as hepatic synthesis. The blue lines represent cholesterol. The release of free fatty acids from chylomicrons is therefore indicated in red, in parallel with the reed color in figure 3. The blue line near CMR is the flux of CMR not directed into the liver. As the reviewer mentions, no numbers are provided for the flux of hepatic VLD-cholesterol secretion. Unfortunately, for this flux numbers are not available, nor for the other lipoprotein fluxes.
New legend of figure 1: “C fluxes in omnivore humans as published by Lütjohann et al [2]. The numbers in italics are obtained by reasonable estimates based on the proportion of chylomicron remnants being extracted by the liver and the proportion of hepatic C synthesis to whole body synthesis. Chylomicron remnants not extracted by the liver are taken up by extrahepatic tissues including macrophages. CM = chylomicrons, CMR = chylomicron remnants, VLDL = very low density lipoprotein, LDL = low density lipoprotein, HDL = high density lipoprotein, FFA = free fatty acid, BA = bile acid. No numbers are available for the lipoprotein fluxes.”
Line 54: It is mentioned here and elsewhere that cholesterol derived from HDL is mainly used for biliary cholesterol secretion. Reference should be added
Response:
The following references have been added:
Nijstad, N.; Gautier, T.; Briand, F.; Rader, D.J.; Tietge, U. Biliary sterol secretion is required for functional in vivo reverse cholesterol transport in mice. J.Gastroenterology. 2011,140, 1043-1051. PMID: 21134376
Lewis, G.F.; Rader, D. New insights into the regulation of HDL metabolism and reverse cholesterol transport. J.Circ. Res. 2005, 24,96, 1221-1232. PMID: 15976321
Line 58; It is stated that “The daily flux ….” Reference should be included
Response:
The following references has been added:
Wilson, M.D.; Rudel, L.L. Review of cholesterol absorption with emphasis on dietary and biliary cholesterol. J. Lipid Res. 1994, 35, 943-955. PMID: 8077852
Lütjohann old reference 2
Lines 86-111: A whole section is devoted on the discussion about egg consumption, which seems out of place and distracts from the point the author is trying to convey. The point regarding the balance of uptake and synthesis and the importance of FAR has been made by describing Kern’s study. The next section is confusing diverts into accompanying foods, LDL and HDL levels. There is a statement suggesting that patients at risk for cardiovascular disease should limit their egg consumption and it is not clear what this is based on. Then the section moves back to FAR.
Response:
The publication of the paper in Nutrients asks for a relevant discussion on dietary aspects. Eggs are important contributors to cholesterol intake and consumption may be generally discouraged. Eggs represent an import commercial product. Therefore, available information on accompanying factors that may increase the risk caused by egg intake (additional fat intake, high fractional absorption rate) or decrease this risk (metabolic HDL effect, reduction of accompanying fat intake) must be adequately considered.
Line 82: ‘this proofs’ – colloquial (this highlights)
Response: Correction has been made
Line 86: ‘nice example’ – colloquial
Response: “Nice” has been replaced by “convincing”
Section 3: This section seems to be predominantly based on data from Lutjohann et al. Are there any confirmatory studies?
Response: The answers to all comments to section 3 are combined below.
Section 3 contains a lot of individual statements with no references and it is not always clear how these statements are linked together. A table or a figure indicating what is known and not known maybe helpful
Line 115: ‘the subjects consumed relatively much cholesterol’ – what is this statement based on?
Line 118-119: ‘We assumed 450 mg/d to be extracted by the liver’ – based on which studies?
Line 122: ‘Generally, a 50% contribution is assumed’ – based on what? Reference?
Line 123: ‘As a compromise here a value of 300 mg/d is applied’ – by who and as a compromise to what?
Lines 126-131: Lots of statements with no references.
Line 129: How does LDL-C influx relate to CMR-C influx?
Response to all comments on section 3.
This section has been added to illustrate that we have clear ideas what fluxes are involved in cholesterol homeostasis in humans but that we lack knowledge on many daily fluxes. The numbers in the text and in figure 1 have been purposely taken from one study involving young healthy German students who participated in a diet study. It would be confusing to try to compare these data with those from other studies performed in other countries in possibly older subjects with different diets and interfering diseases. Also, numbers may have been obtained with different methodologies. The numbers applied in this manuscript are from one study and coherent. Two assumptions have been made since two sources of information were lacking: how much CMR cholesterol is directed to extrahepatic tissues and what is the contribution of hepatic cholesterol synthesis to whole body cholesterol synthesis. I decided to make the assumptions in order to provide a generalized picture of hepatic influxes and effluxes under physiological conditions. Extrahepatic tissues are able to synthesis cholesterol and take up cholesterol from blood via LDL.Therefore, we assumed the majority of CMCR cholesterol (450 from 525 mg/d) to enter the liver. Whole body cholesterol synthesis has been measured in the Lütjohann study [2] indicating a value of 840 mg/d. For humans, no data are available for hepatic cholesterol synthesis. In the cynomolgus monkey, hepatic cholesterol synthesis contributed around 20% of whole body synthesis. In scholar publications a value of 50% is used for humans without references. Both options would lead to an hepatic synthesis of 165 and 420 mg/d. An in between value of 300 mg/d was chosen as a compromise. Despite the clear high uncertainty, this value gives a rough estimate for hepatic synthesis.The LDL-C influx is controlled by the VLDL-C secretion rate. What fraction of the CMR-C influx is taken up in VLDL particles, is unknown. I fully understand the reviewers questions, also about the calculations involved. Therefore, I have clarified some parts of section 3 more clearly and added information requested by another reviewer. Please see manuscript.
Line 190-191: ‘The time frame in which the micelles pass the duodenum and jejunum may be around two hours’ – add reference
Response:
Literature describing transit times in intestinal segments is from the 1960s and 1970s. Unfortunately, the papers cannot be uploaded by PubMed. The information is not of critical importance. Therefore, the sentence has been removed.
Line 196: new paragraph
Response: A new paragraph has been started.
Lines 189-200 and 200-203 seem to state the same point
Response:
Unfortunately, I must disagree with the reviewer. In lines 189-200 I focus totally on intestinal cholesterol absorption. Thereafter, I added the point that the same transporter combination of NPC1L1 and ABCG5/G8 is responsible for biliary secretion of cholesterol. Downregulation of NPC1L1 will lead to a higher biliary cholesterol secretion rate and a lower cholesterol absorption rate, i.e. increased elimination. Downregulation of ABCG5/G8 leads to disturbed biliary cholesterol secretion and a higher cholesterol absorption rate, i.e. reduced elimination.
Line 206: new paragraph
Response: A new paragraph has been started.
Line 214: In health, rephrase
Response: “in health” has been changed into “in healthy subjects”
Figure 2: the figure suggests that absorption is only occurring in the jejunum
Response:
The jejunum may not be the only site of absorption, but at least the major site. In the text, the following sentence have been added: “C is predominantly absorbed in the jejunum [Wilson, PMID: 8077852 ,Sundaram, PMID: 11873098, Glaysher, PMID: 33131909 ]
Wilson, M.D.; Rudel, L.L. Review of cholesterol absorption with emphasis on dietary and biliary cholesterol. J. Lipid Res. 1994, 35, 943-955.PMID: 8077852
Sundaram, A.; Koutkia, P.; Apovian, C.M. Nutritional management of short bowel syndrome in adults.
J, Clin, Gastroenterol. 2002, 34, 207-2020. PMID: 11873098
Glaysher, M.A.; Ward, J.; Aldhwayan, M.; Ruban, A.; Prechtl, C.G.; Fisk, H.L., Chhina, N.; Al-Najim, W.; Smith, C.; Klimowska-Nassar, N.; Johnson, N.; Falaschetti, E.; Goldstone, A.P.; Miras, A.D.; Byrne, J..; Calder, P.C.; Teare, J.P. The effect of a duodenal-jejunal bypass liner on lipid profile and blood concentrations of long chain polyunsaturated fatty acids. Clin. Nutr. 2021, 40,2343-2354. PMID: 33131909 Nutritional management of short bowel syndrome in adults.
Line 231: The LDLr is not the only receptor involved in CMR uptake
Response:
Indeed, the LDL-receptor-related protein (LRP) has been shown to be involved in CMCR uptake. The text and figure 3 have been corrected.
Line 232: ‘Let’s predict’ – colloquial
Response: Let’s predict” has been changed into “it may be predicted”
Line 245: new paragraph and not sure why the HDL comment is inserted here as it has been mentioned twice before. Similarly, it is unclear why ‘C is synthesized as a polymer of acetate units. The rate limiting enzyme involved is 3-hydroxy-3-methylglutaryl coenzyme A reductase (HMG-CoA reductase)’ are mentioned here.
Response:
I thank the reviewer for pointing out confusion due to my statements. The HDL comment has been removed. The sentence that cholesterol is synthesized as a polymer of acetate units has been removed. A new paragraph has started as follows: C is synthesized by a complex enzyme system. The rate limiting enzyme involved is 3-hydroxy-3-methylglutaryl coenzyme A reductase (HMG-CoA reductase).
Line 252: it is unclear what is meant by ‘BA synthesis has its own control’
Response:
It is meant that BA synthesis is controlled by BAs in the enterohepatic circulation. This has been clarified in the text:
“BA synthesis has its own control. In the ileocyte the Farnesoid X receptor (FXR) is activated by BAs which leads to release of fibroblast growth factor 19 (FGF19) which inhibits the rate limiting enzyme in BA synthesis C-7α-hydroxylase.
Figure 4: from the figure it appears that HDL-derived CE is contributing to the TG pool
Response:
No, HDL also contains a limited amount of TG. This is indicated in figure 4.
Line 270: replace ‘otherwise’ with ‘on the other hand’
Response: “Otherwise” has been replaced by “on the other hand”
Lines 280-281: ‘In a later stage, a new method was developed to estimate biliary C secretion in humans’ - Provide more detail on how the method changed. It is unclear how this relates to the next sentence: ‘Also, in humans an undefined external flux added to faecal C excretion’
Response:
The following explanation has been added: “In a later stage, a new method was developed to estimate biliary C secretion in humans since quantitative bile collection is not possible in humans. BA kinetics were measured with stable isotopes as well as the fractional turnover rate of BAs in plasma. These data were transferred to a biliary BA secretion rate. In bile, collected with CCK induced gallbladder contraction, biliary C secretion rate was calculated. Using this procedure also in humans a TICE flux could be demonstrated [27].”
Lines 287-288: ‘Otherwise, for balance studies, biliary C excretion and TICE do not necessarily need to be separated’ – rephrase and explain
Response:
An explanation has been added: “Otherwise, for balance studies, biliary C excretion and TICE do not necessarily need to be separated. In balance studies a total fecal C excretion rate is measured independent of the source. After correction for loss of dietary C, a value remains for loss of biliary C and TCE together.”
Line 291: remove also
Response: “Also” has been removed
Line 292: replace ‘is’ with ‘are’
Response: “Is” has been replaced by “are” and “TG” replaced by TGs
Line 295: ‘Recently, the difference between VLDL-remnants and LDL is not strict anymore’ – colloquial, rephrase and add reference
Response:
A sentence has been added: “Recently, the difference between VLDL-remnants and LDL is not strict anymore. In addition to LDL-C, VLDL remnants C attribute to the risk for cardiovascular disease [Heideman, PMID: 32810544, Balling, PMID: 33272366]
Heidemann, B.E.; Koopal, C.; Bots, M.L; Asselbergs, F.W.; Westerink, J.; Visseren, F.L. The relation between VLDL-cholesterol and risk of cardiovascular events in patients with manifest cardiovascular disease. J.Int. J. Cardiol. 2021, 322, 251-257. PMID: 32810544
Balling, M.; Afzal, S.; Varbo, A.; Langsted, A.; Davey Smith, G.; Nordestgaard, B.G. VLDL Cholesterol Accounts for One-Half of the Risk of Myocardial Infarction Associated With apoB-Containing Lipoproteins.
- Am. Coll. Cardiol. 2020, 76, 2725-2735. PMID: 33272366
Line 302: The more – colloquial (more abundant?)
Response: “The more “ has been replaced by “the more abundant”
Line 303: ‘Independent of size, each particle contains one ApoB-100 unit’ - it is unclear whether this only applies to LDL or also VLDL and VLDL-remnants
Response: This applies also to VLDL particles
Line 315: duplication of line 245
Response:
Line 245 has been removed due to irrelevance at that position. Line 315 is essential as it explains the basis of the isotope methodology.
Section 9: A table listing the different methods with pro’s and con’s would be helpful
Response:
Different methods cover different functionalities (cholesterol absorption, cholesterol synthesis, bile acid synthesis) and different options of using complicated sterol balance and isotope techniques or serum marker measurements. Marker measurements are less accurate, the sterol balance and isotope techniques not suitable in routine patient care. This has been clearly indicated in section 9. In my opinion, a comparison of the experimental techniques appears out of the scope of paper.
Line 352: ‘For C absorption, it may need discussion what flux is important’ is unclear.
Response:
This refers to what is explained in lines 353 and following lines. The fractional absorption rate (FAR) expresses % absorption. Combining this value with the dietary cholesterol intake (mg/d) and biliary secretion rate (mg/d) results in absolute absorption rates (mg/d).
Line 376: The statement: ‘Each of the steps require its own research’ seems at odds with the main message of the manuscript, that cholesterol homeostasis is complex, involves many pathways that are interlinked. They should be studied in the context of each other.
Response:
I fully agree with the reviewer. However, in practice, different scientific groups work at lipoprotein metabolism, hepatic cholesterol metabolism, cholesterol absorption and bile acid metabolism focused on their own targets applying their own techniques.
Line 411: ‘anyway’ – colloquial
The sentence “Anyway , a reduced C absorption rate is readily compensated by increased synthesis.” Has been changed to “ A major drawback of reducing C absorption rate remains the compensating increased synthesis.”
Line 415: it is stated that only statins are available for the treatment of hypercholesterolemia which is not correct. There are other treatment options such as PCSK9 inhibitors.
Response:
It was meant to state that only statins are available to reduce cholesterol synthesis. The following text has been added.
In hypercholesterolemic patients, statin treatment is the first choice to lower serum C. In case of insufficient serum C reduction, ezetimibe treatment is added as a co-treatment. So far, only statins are available as the basic pharmacological tool to reduce C synthesis. For patients that do not tolerate statins, Bempedoic acid can now be prescribed, which blocks adenosine triphosphate-citrate lyase in the liver which is involved in cholesterol production [Pirillo, PMID: 34980867 . When the combined treatment does not achieve sufficient LDL-C reduction, the next step is additional treatment with a proprotein convertase subtilisin/kexin type 9 (PCSK9) inhibitor [Bagepally B, PMID: 34708270] . PCSK9 binds to the LDL-receptor and inhibits the receptor activity. This results in enhanced serum LDL-C levels. Two PCSK9 inhibitors are now on the market: Alirocumab and Evolocumab. Additionally, Inclisiran has been developed, a small interfering RNA that inhibits translation of the protein PCSK9 [Smith, PMID: 35279835].
Pirillo, A.; Catapano, A..L. New insights into the role of bempedoic acid and ezetimibe in the treatment of hypercholesterolemia. Curr. Opin. Endocrinol. Diabetes Obes. 2022, 29, 161-166. PMID: 34980867
Bagepally, B.S.; Sasidharan, A. Incremental net benefit of lipid-lowering therapy with PCSK9 inhibitors: a systematic review and meta-analysis of cost-utility studies. Eur. J. Clin. Pharmacol. 2022, 78, 351-363. PMID: 34708270
Smith, K.W.; White, C.M. Inclisiran: A Novel Small Interfering RNA Drug for Low-Density Lipoprotein Reduction. J. Clin. Pharmacol. 2022, Mar 13. PMID: 35279835
Minor point
Abbreviations not consistently used throughout the manuscript
Response: Use of abbreviations has been checked
Reviewer 2
Whole body cholesterol levels are carefully maintained by balancing input (intestinal absorption and cholesterol synthesis) and elimination (conversion to bile acids and faecal excretion). This is a complex area, involving multiple interlinked pathways with many of the underlying regulatory mechanisms still unknown. The area is hampered further by difficulties in technologies to measure and investigate pathways in humans. Cholesterol metabolism involves the intestine, blood, liver and gallbladder. This review describes each of the steps involved in cholesterol metabolism from uptake to excretion and summarizes the current knowledge of what drives the fluxes between the various steps, paying particular attention to the role of dietary cholesterol.
This manuscript gives a clear overview of the complexity of cholesterol metabolism, what is known and what is unknown in this area. The language, in some parts, is quite colloquial and many statements are general in nature and it is not clear whether they are thoughts of the author or whether they are based on scientific literature.
Main points
Figure 1 is not very clear. Most of the abbreviations are not explained and it is unclear what the thickness of the lines indicates. Why is the FFA line red? What does the blue line near CMR represent? There are no numbers indicated on the line for VLDL synthesis. If the flux for this is unknown this should be indicated.
Response:
I thank the reviewer for pointing out some clear weaknesses in figure 1. All abbreviations are now clarified in the legend. The different thicknesses of the lines indicate the differences in flux sizes due to absorption and diversion of CMR into liver and macrophages as well as hepatic synthesis. The blue lines represent cholesterol. The release of free fatty acids from chylomicrons is therefore indicated in red, in parallel with the reed color in figure 3. The blue line near CMR is the flux of CMR not directed into the liver. As the reviewer mentions, no numbers are provided for the flux of hepatic VLD-cholesterol secretion. Unfortunately, for this flux numbers are not available, nor for the other lipoprotein fluxes.
New legend of figure 1: “C fluxes in omnivore humans as published by Lütjohann et al [2]. The numbers in italics are obtained by reasonable estimates based on the proportion of chylomicron remnants being extracted by the liver and the proportion of hepatic C synthesis to whole body synthesis. Chylomicron remnants not extracted by the liver are taken up by extrahepatic tissues including macrophages. CM = chylomicrons, CMR = chylomicron remnants, VLDL = very low density lipoprotein, LDL = low density lipoprotein, HDL = high density lipoprotein, FFA = free fatty acid, BA = bile acid. No numbers are available for the lipoprotein fluxes.”
Line 54: It is mentioned here and elsewhere that cholesterol derived from HDL is mainly used for biliary cholesterol secretion. Reference should be added
Response:
The following references have been added:
Nijstad, N.; Gautier, T.; Briand, F.; Rader, D.J.; Tietge, U. Biliary sterol secretion is required for functional in vivo reverse cholesterol transport in mice. J.Gastroenterology. 2011,140, 1043-1051. PMID: 21134376
Lewis, G.F.; Rader, D. New insights into the regulation of HDL metabolism and reverse cholesterol transport. J.Circ. Res. 2005, 24,96, 1221-1232. PMID: 15976321
Line 58; It is stated that “The daily flux ….” Reference should be included
Response:
The following references has been added:
Wilson, M.D.; Rudel, L.L. Review of cholesterol absorption with emphasis on dietary and biliary cholesterol. J. Lipid Res. 1994, 35, 943-955. PMID: 8077852
Lütjohann old reference 2
Lines 86-111: A whole section is devoted on the discussion about egg consumption, which seems out of place and distracts from the point the author is trying to convey. The point regarding the balance of uptake and synthesis and the importance of FAR has been made by describing Kern’s study. The next section is confusing diverts into accompanying foods, LDL and HDL levels. There is a statement suggesting that patients at risk for cardiovascular disease should limit their egg consumption and it is not clear what this is based on. Then the section moves back to FAR.
Response:
The publication of the paper in Nutrients asks for a relevant discussion on dietary aspects. Eggs are important contributors to cholesterol intake and consumption may be generally discouraged. Eggs represent an import commercial product. Therefore, available information on accompanying factors that may increase the risk caused by egg intake (additional fat intake, high fractional absorption rate) or decrease this risk (metabolic HDL effect, reduction of accompanying fat intake) must be adequately considered.
Line 82: ‘this proofs’ – colloquial (this highlights)
Response: Correction has been made
Line 86: ‘nice example’ – colloquial
Response: “Nice” has been replaced by “convincing”
Section 3: This section seems to be predominantly based on data from Lutjohann et al. Are there any confirmatory studies?
Response: The answers to all comments to section 3 are combined below.
Section 3 contains a lot of individual statements with no references and it is not always clear how these statements are linked together. A table or a figure indicating what is known and not known maybe helpful
Line 115: ‘the subjects consumed relatively much cholesterol’ – what is this statement based on?
Line 118-119: ‘We assumed 450 mg/d to be extracted by the liver’ – based on which studies?
Line 122: ‘Generally, a 50% contribution is assumed’ – based on what? Reference?
Line 123: ‘As a compromise here a value of 300 mg/d is applied’ – by who and as a compromise to what?
Lines 126-131: Lots of statements with no references.
Line 129: How does LDL-C influx relate to CMR-C influx?
Response to all comments on section 3.
This section has been added to illustrate that we have clear ideas what fluxes are involved in cholesterol homeostasis in humans but that we lack knowledge on many daily fluxes. The numbers in the text and in figure 1 have been purposely taken from one study involving young healthy German students who participated in a diet study. It would be confusing to try to compare these data with those from other studies performed in other countries in possibly older subjects with different diets and interfering diseases. Also, numbers may have been obtained with different methodologies. The numbers applied in this manuscript are from one study and coherent. Two assumptions have been made since two sources of information were lacking: how much CMR cholesterol is directed to extrahepatic tissues and what is the contribution of hepatic cholesterol synthesis to whole body cholesterol synthesis. I decided to make the assumptions in order to provide a generalized picture of hepatic influxes and effluxes under physiological conditions. Extrahepatic tissues are able to synthesis cholesterol and take up cholesterol from blood via LDL.Therefore, we assumed the majority of CMCR cholesterol (450 from 525 mg/d) to enter the liver. Whole body cholesterol synthesis has been measured in the Lütjohann study [2] indicating a value of 840 mg/d. For humans, no data are available for hepatic cholesterol synthesis. In the cynomolgus monkey, hepatic cholesterol synthesis contributed around 20% of whole body synthesis. In scholar publications a value of 50% is used for humans without references. Both options would lead to an hepatic synthesis of 165 and 420 mg/d. An in between value of 300 mg/d was chosen as a compromise. Despite the clear high uncertainty, this value gives a rough estimate for hepatic synthesis.The LDL-C influx is controlled by the VLDL-C secretion rate. What fraction of the CMR-C influx is taken up in VLDL particles, is unknown. I fully understand the reviewers questions, also about the calculations involved. Therefore, I have clarified some parts of section 3 more clearly and added information requested by another reviewer. Please see manuscript.
Line 190-191: ‘The time frame in which the micelles pass the duodenum and jejunum may be around two hours’ – add reference
Response:
Literature describing transit times in intestinal segments is from the 1960s and 1970s. Unfortunately, the papers cannot be uploaded by PubMed. The information is not of critical importance. Therefore, the sentence has been removed.
Line 196: new paragraph
Response: A new paragraph has been started.
Lines 189-200 and 200-203 seem to state the same point
Response:
Unfortunately, I must disagree with the reviewer. In lines 189-200 I focus totally on intestinal cholesterol absorption. Thereafter, I added the point that the same transporter combination of NPC1L1 and ABCG5/G8 is responsible for biliary secretion of cholesterol. Downregulation of NPC1L1 will lead to a higher biliary cholesterol secretion rate and a lower cholesterol absorption rate, i.e. increased elimination. Downregulation of ABCG5/G8 leads to disturbed biliary cholesterol secretion and a higher cholesterol absorption rate, i.e. reduced elimination.
Line 206: new paragraph
Response: A new paragraph has been started.
Line 214: In health, rephrase
Response: “in health” has been changed into “in healthy subjects”
Figure 2: the figure suggests that absorption is only occurring in the jejunum
Response:
The jejunum may not be the only site of absorption, but at least the major site. In the text, the following sentence have been added: “C is predominantly absorbed in the jejunum [Wilson, PMID: 8077852 ,Sundaram, PMID: 11873098, Glaysher, PMID: 33131909 ]
Wilson, M.D.; Rudel, L.L. Review of cholesterol absorption with emphasis on dietary and biliary cholesterol. J. Lipid Res. 1994, 35, 943-955.PMID: 8077852
Sundaram, A.; Koutkia, P.; Apovian, C.M. Nutritional management of short bowel syndrome in adults.
J, Clin, Gastroenterol. 2002, 34, 207-2020. PMID: 11873098
Glaysher, M.A.; Ward, J.; Aldhwayan, M.; Ruban, A.; Prechtl, C.G.; Fisk, H.L., Chhina, N.; Al-Najim, W.; Smith, C.; Klimowska-Nassar, N.; Johnson, N.; Falaschetti, E.; Goldstone, A.P.; Miras, A.D.; Byrne, J..; Calder, P.C.; Teare, J.P. The effect of a duodenal-jejunal bypass liner on lipid profile and blood concentrations of long chain polyunsaturated fatty acids. Clin. Nutr. 2021, 40,2343-2354. PMID: 33131909 Nutritional management of short bowel syndrome in adults.
Line 231: The LDLr is not the only receptor involved in CMR uptake
Response:
Indeed, the LDL-receptor-related protein (LRP) has been shown to be involved in CMCR uptake. The text and figure 3 have been corrected.
Line 232: ‘Let’s predict’ – colloquial
Response: Let’s predict” has been changed into “it may be predicted”
Line 245: new paragraph and not sure why the HDL comment is inserted here as it has been mentioned twice before. Similarly, it is unclear why ‘C is synthesized as a polymer of acetate units. The rate limiting enzyme involved is 3-hydroxy-3-methylglutaryl coenzyme A reductase (HMG-CoA reductase)’ are mentioned here.
Response:
I thank the reviewer for pointing out confusion due to my statements. The HDL comment has been removed. The sentence that cholesterol is synthesized as a polymer of acetate units has been removed. A new paragraph has started as follows: C is synthesized by a complex enzyme system. The rate limiting enzyme involved is 3-hydroxy-3-methylglutaryl coenzyme A reductase (HMG-CoA reductase).
Line 252: it is unclear what is meant by ‘BA synthesis has its own control’
Response:
It is meant that BA synthesis is controlled by BAs in the enterohepatic circulation. This has been clarified in the text:
“BA synthesis has its own control. In the ileocyte the Farnesoid X receptor (FXR) is activated by BAs which leads to release of fibroblast growth factor 19 (FGF19) which inhibits the rate limiting enzyme in BA synthesis C-7α-hydroxylase.
Figure 4: from the figure it appears that HDL-derived CE is contributing to the TG pool
Response:
No, HDL also contains a limited amount of TG. This is indicated in figure 4.
Line 270: replace ‘otherwise’ with ‘on the other hand’
Response: “Otherwise” has been replaced by “on the other hand”
Lines 280-281: ‘In a later stage, a new method was developed to estimate biliary C secretion in humans’ - Provide more detail on how the method changed. It is unclear how this relates to the next sentence: ‘Also, in humans an undefined external flux added to faecal C excretion’
Response:
The following explanation has been added: “In a later stage, a new method was developed to estimate biliary C secretion in humans since quantitative bile collection is not possible in humans. BA kinetics were measured with stable isotopes as well as the fractional turnover rate of BAs in plasma. These data were transferred to a biliary BA secretion rate. In bile, collected with CCK induced gallbladder contraction, biliary C secretion rate was calculated. Using this procedure also in humans a TICE flux could be demonstrated [27].”
Lines 287-288: ‘Otherwise, for balance studies, biliary C excretion and TICE do not necessarily need to be separated’ – rephrase and explain
Response:
An explanation has been added: “Otherwise, for balance studies, biliary C excretion and TICE do not necessarily need to be separated. In balance studies a total fecal C excretion rate is measured independent of the source. After correction for loss of dietary C, a value remains for loss of biliary C and TCE together.”
Line 291: remove also
Response: “Also” has been removed
Line 292: replace ‘is’ with ‘are’
Response: “Is” has been replaced by “are” and “TG” replaced by TGs
Line 295: ‘Recently, the difference between VLDL-remnants and LDL is not strict anymore’ – colloquial, rephrase and add reference
Response:
A sentence has been added: “Recently, the difference between VLDL-remnants and LDL is not strict anymore. In addition to LDL-C, VLDL remnants C attribute to the risk for cardiovascular disease [Heideman, PMID: 32810544, Balling, PMID: 33272366]
Heidemann, B.E.; Koopal, C.; Bots, M.L; Asselbergs, F.W.; Westerink, J.; Visseren, F.L. The relation between VLDL-cholesterol and risk of cardiovascular events in patients with manifest cardiovascular disease. J.Int. J. Cardiol. 2021, 322, 251-257. PMID: 32810544
Balling, M.; Afzal, S.; Varbo, A.; Langsted, A.; Davey Smith, G.; Nordestgaard, B.G. VLDL Cholesterol Accounts for One-Half of the Risk of Myocardial Infarction Associated With apoB-Containing Lipoproteins.
- Am. Coll. Cardiol. 2020, 76, 2725-2735. PMID: 33272366
Line 302: The more – colloquial (more abundant?)
Response: “The more “ has been replaced by “the more abundant”
Line 303: ‘Independent of size, each particle contains one ApoB-100 unit’ - it is unclear whether this only applies to LDL or also VLDL and VLDL-remnants
Response: This applies also to VLDL particles
Line 315: duplication of line 245
Response:
Line 245 has been removed due to irrelevance at that position. Line 315 is essential as it explains the basis of the isotope methodology.
Section 9: A table listing the different methods with pro’s and con’s would be helpful
Response:
Different methods cover different functionalities (cholesterol absorption, cholesterol synthesis, bile acid synthesis) and different options of using complicated sterol balance and isotope techniques or serum marker measurements. Marker measurements are less accurate, the sterol balance and isotope techniques not suitable in routine patient care. This has been clearly indicated in section 9. In my opinion, a comparison of the experimental techniques appears out of the scope of paper.
Line 352: ‘For C absorption, it may need discussion what flux is important’ is unclear.
Response:
This refers to what is explained in lines 353 and following lines. The fractional absorption rate (FAR) expresses % absorption. Combining this value with the dietary cholesterol intake (mg/d) and biliary secretion rate (mg/d) results in absolute absorption rates (mg/d).
Line 376: The statement: ‘Each of the steps require its own research’ seems at odds with the main message of the manuscript, that cholesterol homeostasis is complex, involves many pathways that are interlinked. They should be studied in the context of each other.
Response:
I fully agree with the reviewer. However, in practice, different scientific groups work at lipoprotein metabolism, hepatic cholesterol metabolism, cholesterol absorption and bile acid metabolism focused on their own targets applying their own techniques.
Line 411: ‘anyway’ – colloquial
The sentence “Anyway , a reduced C absorption rate is readily compensated by increased synthesis.” Has been changed to “ A major drawback of reducing C absorption rate remains the compensating increased synthesis.”
Line 415: it is stated that only statins are available for the treatment of hypercholesterolemia which is not correct. There are other treatment options such as PCSK9 inhibitors.
Response:
It was meant to state that only statins are available to reduce cholesterol synthesis. The following text has been added.
In hypercholesterolemic patients, statin treatment is the first choice to lower serum C. In case of insufficient serum C reduction, ezetimibe treatment is added as a co-treatment. So far, only statins are available as the basic pharmacological tool to reduce C synthesis. For patients that do not tolerate statins, Bempedoic acid can now be prescribed, which blocks adenosine triphosphate-citrate lyase in the liver which is involved in cholesterol production [Pirillo, PMID: 34980867 . When the combined treatment does not achieve sufficient LDL-C reduction, the next step is additional treatment with a proprotein convertase subtilisin/kexin type 9 (PCSK9) inhibitor [Bagepally B, PMID: 34708270] . PCSK9 binds to the LDL-receptor and inhibits the receptor activity. This results in enhanced serum LDL-C levels. Two PCSK9 inhibitors are now on the market: Alirocumab and Evolocumab. Additionally, Inclisiran has been developed, a small interfering RNA that inhibits translation of the protein PCSK9 [Smith, PMID: 35279835].
Pirillo, A.; Catapano, A..L. New insights into the role of bempedoic acid and ezetimibe in the treatment of hypercholesterolemia. Curr. Opin. Endocrinol. Diabetes Obes. 2022, 29, 161-166. PMID: 34980867
Bagepally, B.S.; Sasidharan, A. Incremental net benefit of lipid-lowering therapy with PCSK9 inhibitors: a systematic review and meta-analysis of cost-utility studies. Eur. J. Clin. Pharmacol. 2022, 78, 351-363. PMID: 34708270
Smith, K.W.; White, C.M. Inclisiran: A Novel Small Interfering RNA Drug for Low-Density Lipoprotein Reduction. J. Clin. Pharmacol. 2022, Mar 13. PMID: 35279835
Minor point
Abbreviations not consistently used throughout the manuscript
Response: Use of abbreviations has been checked
Reviewer 3 Report
This Review is a timely attempt to integrate the knowledge on cholesterol homeostasis and its relation with dietary and endogenous cholesterol fluxes. The Review is of particular interest as it describes the potential parallel connections of cholesterol, bile acids, and triglyceride fluxes through co-metabolism and co-transport in the enterohepatic circulation.
The Review may benefit from some additional updates and discussion to better report the overall fluxes and hypothesis. In particular on the following two points:
- Recent findings support a role for gut bacterial metabolism in modulating host cholesterol levels (PMID: 32308619). Coprostanol is not absorbed efficiently in the gastrointestinal tract. An inverse relationship between serum cholesterol levels and the fecal ratio of cholesterol to coprostanol or cholesterol-metabolizing bacterial genes species has been observed (PMID: 32544460; PMID: 34576776; PMID: 31189655). The catabolism through cholesterol reduction into coprostanol is mentioned in the introduction but not discussed hereafter. It is a poorly characterized pathway but could be introduced in section 3, “Known and unknown cholesterol fluxes in Human.
- Similarly, cholesterol fluxes related to epithelial cell desquamation/cell renewal and steroid hormones could be reported.
For consistency, the Authors should match the average size of cholesterol fluxes described in the text with those shown in figure 1.
- Line 118: how is it calculated 525mg/d?
- Line 120: in the Paper of Lutjohann et al., the bile synthesis rate is not reported. 420 mg/day corresponds to bile acid excretion.
Line 127: 620mg is reported, but the figure shows 450+300=750.
Line 245-246: a reference is needed.
Author Response
Reviewer 3
This Review is a timely attempt to integrate the knowledge on cholesterol homeostasis and its relation with dietary and endogenous cholesterol fluxes. The Review is of particular interest as it describes the potential parallel connections of cholesterol, bile acids, and triglyceride fluxes through co-metabolism and co-transport in the enterohepatic circulation.
Response:
I am very grateful for the reviewers positive reply and for the detailed set of comments provide by the reviewer. They will clearly lead to a strong improvement of this paper.
The Review may benefit from some additional updates and discussion to better report the overall fluxes and hypothesis. In particular on the following two points:
- Recent findings support a role for gut bacterial metabolism in modulating host cholesterol levels (PMID: 32308619). Coprostanol is not absorbed efficiently in the gastrointestinal tract. An inverse relationship between serum cholesterol levels and the fecal ratio of cholesterol to coprostanol or cholesterol-metabolizing bacterial genes species has been observed (PMID: 32544460; PMID: 34576776; PMID: 31189655). The catabolism through cholesterol reduction into coprostanol is mentioned in the introduction but not discussed hereafter. It is a poorly characterized pathway but could be introduced in section 3, “Known and unknown cholesterol fluxes in Human.
Response:
I thank the reviewer for the suggestions and added the gut metabolism aspects at the end of section 3:
Another largely unexplored aspect of C metabolism is the bacterial metabolism of C that escapes small intestinal absorption. Based on the fluxes presented in figure 1 the daily flux of malabsorbed C averages 525 mg/d. The major metabolic metabolite is coprostanol [Juste, PMID: 34576776 ]. The conversion rate, expressed as the coprostanol/cholesterol ratio in feces of healthy subjects is high (high converters), but may be low in a minor group of subjects (low converters). Only a small number of bacterial strains have been identified that is able to produce coprostanol from cholesterol.This process has been shown to modify host C levels in serum [Vilette, PMID: 32308619]. The ratio coprostanol/C in feces appears inversely related to the serum C concentration [Kenny, PMID: 32544460 ]. Diet factors may intervene with the colonic bacterial C metabolism as has been documented for milk polar lipids reducing serum C concentration [Fors, PMID: 31189655].
- Similarly, cholesterol fluxes related to epithelial cell desquamation/cell renewal and steroid hormones could be reported.
Response:
A difficult aspect of writing a review paper is to determine the exact focus of the theme. Cholesterol metabolism is involved in many physiological and pathological processes and many processes are involved in cholesterol metabolism. An important aspect in writing this review is, that it is meant to serve as an introductory review for the special Nutrients issue “From dietary cholesterol to blood cholesterol”. This issue tries to link dietary cholesterol to blood cholesterol and as a consequence to cardiovascular disease. Therefore, the title “From dietary cholesterol to blood cholesterol, physiological lipid fluxes and cholesterol homeostasis” was chosen. Cholesterol fluxes related to epithelial cell desquamation/cell renewal and steroid hormones appears to be outside the scope of the paper.
For consistency, the Authors should match the average size of cholesterol fluxes described in the text with those shown in figure 1.
Response:
The data have been checked and corrected in the text of section 3. In section 3, the data in figure 1 are now better explained. For further explanation see the answers to the next comments.
- Line 118: how is it calculated 525mg/d?
Response:
The mean dietary C influx was 350 mg/d, the mean biliary C influx twice as much being 700 mg/d. Applying the mean C fractional absorption rate of 50%, the calculated flux of absorbed C then equals 525 mg/d. The text has been changed as follows: “With a mean daily C intake of 350 mg/d, the subjects had a relatively C consumption, when compared to the normal intake of 100 to 400 mg/d. The data show that the daily biliary C flux is twice the dietary C flux (700 mg/d). The total daily intestinal C flux is then 1050 mg/d. Based on the mean 50% FAR, the daily total flux of absorbed C is on average 525 mg/d. The majority will be taken up by the liver, the rest enters extrahepatic tissues including macrophages..”
- Line 120: in the Paper of Lutjohann et al., the bile synthesis rate is not reported. 420 mg/day corresponds to bile acid excretion.
Response:
In the kinetic model of bile acid metabolism, the fecal bile acid excretion is equally compensated by bile acid synthesis. Thus fecal excretion can be directly translated into synthesis.
Line 127: 620mg is reported, but the figure shows 450+300=750.
Response;
I thank the reviewer for the critical reading and finding this mistake. The number has been corrected.
Line 245-246: a reference is needed.
Response:
The following references have been added:
Nijstad, N.; Gautier, T.; Briand, F.; Rader, D.J.; Tietge, U. Biliary sterol secretion is required for functional in vivo reverse cholesterol transport in mice. J.Gastroenterology. 2011,140, 1043-1051. PMID: 21134376
Lewis, G.F.; Rader, D. New insights into the regulation of HDL metabolism and reverse cholesterol transport. J.Circ. Res. 2005, 24,96, 1221-1232. PMID: 15976321
Round 2
Reviewer 2 Report
None. The author has appropriately addressed all issues raised. The changes have improved the manuscript and it is now suitable for publication